# Cognitive Reserve in Early Manifest Huntington Disease Patients: Leisure Time Is Associated with Lower Cognitive and Functional Impairment

**DOI:** 10.3390/jpm12010036

**Published:** 2022-01-03

**Authors:** Simone Migliore, Giulia D’Aurizio, Eugenia Scaricamazza, Sabrina Maffi, Consuelo Ceccarelli, Giovanni Ristori, Silvia Romano, Anna Castaldo, Mario Fichera, Giuseppe Curcio, Ferdinando Squitieri

**Affiliations:** 1Huntington and Rare Diseases Unit, Fondazione IRCCS Casa Sollievo Della Sofferenza Hospital, 71013 San Giovanni Rotondo, Italy; sim.migliore@gmail.com (S.M.); eugenia138@hotmail.com (E.S.); sabrina.maffi@gmail.com (S.M.); 2Department of Biotechnological and Applied Clinical Sciences, University of L’Aquila, 67100 L’Aquila, Italy; giulia.daurizio@gmail.com (G.D.); giuseppe.curcio@univaq.it (G.C.); 3Italian League for Research on Huntington and Related Diseases Foundation, 00185 Rome, Italy; consuelo.ceccarelli@lirh.it; 4Centre for Experimental Neurological Therapies, Department of Neuroscience, Mental Health and Sensory Organs, Faculty of Medicine and Psychology, S. Andrea Hospital, Sapienza University, 00189 Rome, Italy; giovanni.ristori@uniroma.it (G.R.); silvia.romano@uniroma1.it (S.R.); 5Department of Medical Genetics and Neurogenetics, Fondazione IRCCS Istituto Neurologico Carlo Besta, 20133 Milan, Italy; anna.Castaldo@istituto-besta.it (A.C.); mario.fichera@istituto-besta.it (M.F.)

**Keywords:** cognitive engagement, neurodegenerative, executive functions, cognition

## Abstract

We focused on Cognitive Reserve (CR) in patients with early Huntington Disease (HD) and investigated whether clinical outcomes might be influenced by lifetime intellectual enrichment over time. CR was evaluated by means of the Cognitive Reserve Index questionnaire (CRIq), an internationally validated scale which includes three sections: education, working activity, and leisure time. The clinical HD variables were quantified at three different time points (baseline-t0, 1 year follow up-t1 and 2 years follow up-t2) as per the Unified Huntington’s Disease Rating Scale (UHDRS), an internationally standardized and validated scale including motor, cognitive, functional and behavioral assays. Our sample consisted of 75 early manifest patients, withclinical stage scored according to the Total Functional Capacity (TFC) scale. Our correlational analysis highlighted a significant inverse association between CRIq leisure time (CRIq_LA) and longitudinal functional impairment (namely, the differential TFC score between t2 and t0 or ΔTFC) (*p* < 0.05), and the multidimensional progression of HD as measured by the composite UHDRS (cUHDRS, *p* < 0.01). CRIq_LA was significantly and positively associated with better cognitive performances at all time points (*p* < 0.05). Our results suggest that higher is the CRIq_LA, milder is the progression of HD in terms of functional, multidimensional and cognitive outcome.

## 1. Introduction

Huntington disease (HD) is a hereditary neurodegenerative disease manifesting with neurological and mental symptoms which may begin at any age, mainly in adulthood. The gene mutation is a CAG triplet expansion that is detectable by a molecular test during the presymptomatic lifetime [1]. Usually, the genetic test is associated with specific psychological and genetic counselling [2]. Neurological features include movement disorders, among which chorea is the landmark, and progressive cognitive decline starting with early emotion recognition deficit [3], spatial memory problems [4], impaired attention functions [5], and finally overt dementia in the later stage of the disease.

According to the Cognitive Reserve (CR) hypothesis, clinical expression of brain diseases might be attenuated among individuals with higher education or active lifestyles [6]. People with greater intellectual enrichment are able to withstand more severe neuropathology before developing cognitive impairment or dementia. Two models of reserve have been described so far: the passive, or brain reserve model and the active, or CR model. The former hypothesizes that each individual disposes of a certain amount of brain capacity (determined by brain volume, number of neurons, etc.) to withstand the development of neurologic diseases [4]. In contrast, the latter speculates that the nervous system might actively cope with brain damage by using more efficient pre-existing neural networks or recruiting additional brain regions [6,7,8]. The passive and active models of CR can be modulated by psychosocial factors. Intellectual enrichment is generally evaluated in terms of educational attainment, vocabulary knowledge, and occupation level [9,10], and many studies have shown that premorbid engagement in leisure activities might increase CR and mitigate longitudinal cognitive impairment [11,12].

The CR hypothesis has been well-supported by studies involving various neurological conditions, such as Alzheimer’s disease (AD) [12], Parkinson’s disease (PD) [13], Multiple Sclerosis (MS) [14], stroke [15], traumatic brain injury [16], and age-related white matter ischemic changes [17].

On the contrary, little is known about CR in HD. While research has begun to disclose the biological factors impacting HD development [18], the effect of environmental factors on the onset, progression and severity of the disease is far less clear. Studies in animal models of HD hint that enriched environments may delay motor onset [19] and significantly affect cognitive impairment [20,21]. Moreover, studies on HD patients suggest that an active lifestyle and increased educational level have a beneficial effect on both clinical symptoms and brain activation in HD [22,23,24,25,26]. Specifically, HD subjects with higher cognitive reserve showed (a) slower decline in specific cognitive domains such as cognitive flexibility, working memory and inhibitory control; (b) slower volume loss in specific brain regions (i.e., putamen, left precuneus and the bilateral caudate), and (c) reinforced connectivity in subcortical and cortical brain structures (i.e., anterior cingulate cortex and left angular gyrus) [22,23,24].

Altogether, these data support the hypothesis that neuroplasticity is an intrinsic property of the living brain that enables it to reorganize its structure, function and networks in response to internal/external changes (i.e., brain pathologies or physical exercise) and may represent a potential protective factor in neurodegeneration [27]. Regional brain overactivity, which might represent a compensatory process in response to neurodegeneration [18,28], has been observed in both the presymptomatic and early stages of HD.

Our study aims to evaluate CR in patients with early manifest HD by investigating whether lifetime intellectual enrichment may be beneficial to cognitive and clinical performance. For this purpose, we used an international validated questionnaire, the Cognitive Reserve Index questionnaire (CRIq), which assesses the acquired lifetime CR in three different areas: education, working activities, and leisure time activities.

## 2. Materials and Methods

### 2.1. Subjects

Starting from a sample of 177 early manifesting patients with at least three yearly clinical examinations, we selected 75 patients based on the following inclusion criteria: (1) motor age at onset above 20 years; (2) Unified Huntington Disease Rating Scale (UHDRS)—Total Motor Score (TMS) > 10 and Diagnostic Confidence Level (DCL) equal to 4 at baseline; (3) UHDRS Total Function Capacity (TFC) > 6 at baseline; (4) no history of neurological conditions other than HD, substance dependence, or developmental disorder affecting cognition at baseline assessment; and (5) no severe psychiatric manifestations. A full patient description is reported in Table 1. HD was genetically confirmed in all cases, with gene expansions being greater than ≥40 CAG repeats. This cohort was recruited by three ENROLL-HD Centres in Italy, e.g., the LIRH Foundation (the coordinator site, Rome), IRCCS Istituto Neurologico Carlo Besta (Milan); and Sant’ Andrea University Hospital (Rome). ENROLL-HD is a large worldwide research platform for observational studies on HD. HD family participants are assessed yearly by standardized medical, motor, behavioral, functional and cognitive measures [29]. All raters are required to be annually certified before performing ENROLL-HD clinical assays.

Our study conforms with the World Medical Association Declaration of Helsinki. It received approval by the Institutional Review Board of the coordinator site (LIRH Foundation, prot. number 102/14, approved in date 28 May 2014); all participants signed a declaration of informed consent.

### 2.2. Clinical Measures

All patients were assessed by health professionals with expertise in HD in accordance with ENROLL-HD guidelines [29]. Evaluations were performed at baseline (time 0; t0) and repeated at 1-year (time 1; t1) and 2-year (time 2; t2) follow-ups by means of the UHDRS [30], which includes motor, cognitive and functional domains. The TMS is a standardized and validated tool for rating motor impairment consisting of 31 items; each item can be scored from 0 (indicating normal performance) to 4 (indicating the most severe impairment) [30]. The TFC is a standardized and validated HD assay of overall functional capacity considering the following domains: occupation, financial management, domestic chores, and daily life activities, with a total score ranging from 13 (normal function) to 0 (complete loss of function) [30,31].

We performed a comprehensive cognitive evaluation including: (1) Categorical Verbal Fluency Test (VFT); (2) Stroop Color Reading Test (SCR); (3) Stroop Word Reading (SWR); (4) Symbol Digit Modalities Test (SDMT) in the written response format; and (5) Mini Mental State Examination (MMSE), according to ENROLL-HD protocol. We calculated the composite UHDRS (cUHDRS), a multidimensional measure of motor (TMS), cognitive (SDMT and SWR) and functional (TFC) outcome, recently proposed as a sensitive tool for monitoring clinical progression in early HD [32].

### 2.3. Cognitive Reserve

CR was assessed by means of the CRIq, an internationally validated instrument that is currently available in ten different languages. The questionnaire is composed of twenty items grouped into three section subscores: (1) years of formal and informal education (CRIq_Edu); (2) working activity experience according to degree of responsibility and cognitive demands (CRIq_WA), with five different levels of working activities available dealing with the degree of intellectual involvement and personal responsibility; and (3) leisure activity, calculated based on the frequency of several activities such as reading, housekeeping, driving, hobbies, travelling etc., on a weekly, monthly, yearly, or stable basis (CRIq_LA) [33]. The total score (CRIq_Tot) and the three subscores (CRIq_Edu, CRIq_WA, CRIq_LA) were considered in our analysis. Experienced health professionals conducted semi-structured interviews to gather pertinent information. The CRIq instructions and the Excel file for automatic calculation of subscores are available at https://dpg.unipd.it/en/criq (accessed on 15 January 2019). CRIq is an efficient and reliable tool for measuring CR [33]; specifically, it is short and easy to administer and thus easily to be included in standard assessments.

### 2.4. Statistical Analysis

First, we conducted an exploratory Pearson’s correlation analysis in order to detect potential relationships between the CRIq scores (CRIq_Tot, CRIq_Edu, CRIq_WA, CRIq_LA) and the clinical and genetic variables under investigation (Age at Onset (AO), expanded CAG repeats, ∆TMS (the difference between the TMS score at t2 and t0), ∆TFC, cUHDRS, cognitive variables (Categorical Verbal Fluency Test, correct response; Stroop Color Reading, correct responses; Stroop Word Reading, correct responses; Symbol Digit Modalities Test, correct responses), and Mini Mental State Examination). The correlation analysis was performed for all time points (t0, t1, and t2). In addition, we investigated the correlation between the mutation size (i.e., expanded CAG repeat number) with each clinical variable.

Patients were then split into two groups depending on whether the CRIq scores obtained were normal or impaired. CRIq subscores were considered pathological when below 1 Standard Deviation (SD) from the control mean; for more details, see [33].

All dependent variables were submitted into a one-way ANOVA comparing the performance of patients with normal and impaired CRIq subscores.

Alpha level was conventionally fixed to ≤0.05. All statistical analyses were performed using IBM SPSS Statistics for Macintosh, version 25.0 (IBM Corp., Armonk, NY, USA).

## 3. Results

### 3.1. Patients’ Characteristics

Our cohort (n = 75) included 28 females and 47 males. The demographic characteristics of the sample included age at baseline of 47.2 ± 12.5 years (range 27–78 years), education level of 11.65 ± 4.6 years (range 8–18 years), age at motor onset 47.5 ± 12.3 years (range 22–70 years), and expanded CAG repeat number of 43.7 ± 2.3 (range 40–49 CAG repeats). The clinical and demographic characteristics are summarized in Table 1. Genetic characteristics with mutation size, including the expanded CAG repeat number, are reported in Appendix A.

### 3.2. Correlation between CR, Clinical and Cognitive Scores

The preliminary Pearson’s analysis highlighted various significant associations between CRIq scores, in particular the CRI_LA subscore, and the clinical/cognitive variables (Figure 1 and Figure 2). Specifically, among the clinical variables, significant positive correlations were found between CRIq_LA and ∆TFC (*p* = 0.037; *r* = 0.242), cUHDRS_t0 (*p* = 0.002; *r* = 0.349), cUHDRS_t1 (*p* = 0.004; *r* = 0.332) and cUHDRS_t2 (*p* = 0.003; *r* = 0.344) (Figure 1, Table 2).

With regard to the cognitive variables, at baseline the analysis showed significant positive correlations between CRIq_LA and SDMT ((*p* = 0.001; *r* = 0.392), VFT (*p* = 0.036, *r* = 0.240), SCR (*p* = 0.015, *r* = 0.280), SWR (*p* = 0.002, *r* = 0.348) and MMSE (*p* = 0.003, *r* = 0.340) (see Figure 2 and Table 3 (a)). We found a significant positive correlation between CRIq_Tot and SDMT (*p* < 0.039; *r* = 0.239) and MMSE (*p* = 0.012, *r* = 0.294) as well (see Table 3 (a)).

At t1, the analysis showed significant positive correlations between CRIq_LA and SDMT (*p* = 0.002); *r* = 0.359), VFT (*p* = 0.022, *r* = 0.265), SCR (*p* = 0.01, *r* = −0.294), SWR (*p* = 0.004, *r* = 0.330) and MMSE (*p* < 0.001, *r* = 0.396; Figure 2 and Table 3(b)) scores. We found a significant positive correlation between CRIq_Tot and MMSE (*p* = 0.003, *r* = 0.337) as well (see Table 3 (b)).

Finally, at t2 the analysis showed significant positive correlations between CRIq_LA and SDMT (*p* = 0.002; *r* = 0.354), VFT (*p* = 0.029, *r* = 0.252), SCR (*p* = 0.028, *r* = 0.253), SWR (*p* = 0.001, *r* = 0.374) and MMSE (*p* < 0.001, *r* = 0.404) (Figure 2 and Table 3 (c)). We again found a significant positive correlation between CRIq_Tot and MMSE (*p* = 0.005, *r* = 0.326; see Table 3 (c)).

The expanded CAG repeat number showed a significant negative correlation with CRIq_Edu (*p* = 0.002, *r* = −0.348), CRIq_WA (*p* = 0.005, *r* = −0.320) and CRIq_Tot (*p* = 0.001, *r* = −0.372; Appendix A). The mutation size did not affect the relationship of CRIq_LA (Appendix A) with clinical (∆TMS, ∆TFC, cUHDRS; Appendix A) or cognitive variables (SDMT, VFT, SCR, SWR and MMSE; Appendix A).

### 3.3. Comparison between Groups with Impaired and Normal Cognitive Reserve Indices

Patients were split into two groups depending on the totalization of a normal or impaired score at CRI assessment; hence a one-way ANOVA comparison was performed.

CRIq Leisure Activity: Among the clinical variables, we found a significant Group effect (F_1,73_ = 5.94; *p* = 0.017) with regard to the ∆TFC, which resulted higher in the low CRIq_LA group (−2.08 ± 0.28) in comparison to the high CRIq_LA group (−1.2 ± 0.21). Moreover, the one-way ANOVA showed a significant group effect on cUHDRS at t0 (F_1,73_ = 11.29; *p* = 0.001), t1 (F_1,73_ = 13.87; *p* = 0.0001) and t2 (F_1,73_ = 11.16; *p* = 0.0001), indicating a more pronounced multidimensional progression in the low CRIq_LA than in the high CRIq_LA group (t0: 54.34 ± 8.23 vs. 92.28 ± 6.69) (t1: 41.09 ± 8.41 vs. 85.48 ± 7.14; t2: 30.09 ± 9.64 vs. 75.5 ± 8.13). No significant differences emerged for ∆TMS (see Table 4 for further details).

Regarding cognitive variables, a significant group effect was observed, showing globally poorer performances in the group with low CRIq_LA compared to the high CRIq_LA group at baseline. The detailed statistical results are listed as follows: MMSE (F_1,70_ = 5.74; *p* = 0.019; 25.34 ± 0.60 vs. 26.98 ± 0.37), SDMT (F_1,73_ = 13.2; *p* = 0.001; 17.54 ± 1.70 vs. 28.58 ± 1.92), VFT (F_1,73_ = 6.03; *p* = 0.016; 11.87 ± 0.84 vs 14.94 ± 0.75), SCR (F_1,73_ = 7.61; *p* = 0.007; 41.41 ± 3.19 vs. 52.88 ± 2.42) and SWR (F_1,73_ = 8.02; *p* = 0.006; 58.75 ± 4.39 vs. 75.02 ± 3.35; see Table 4 for further details).

The one-way ANOVA confirmed a significant Group effect on MMSE (F_1,73_ = 16.1; *p* = 0.0001; 23.62 ± 0.74 vs. 26.78 ± 0.41), SDMT (F_1,71_ = 15.1; *p* = 0.0001; 16.04 ± 1.84 vs. 28.06 ± 1.91), VFT (F_1,73_ = 9.72; *p* = 0.003; 10.7 ± 1 vs. 14.94 ± 0.8), SCR (F_1,73_ = 6.54; *p* = 0.013) (39.25 ± 2.71 vs. 50.17 ± 2.63) and SWR (F_1,73_ = 10.22; *p* = 0.002; 54.16 ± 3.92 vs. 72.6 ± 3.49) at 1 year follow-up, indicating poorer performance in the impaired CRIq_LA group with respect to the normal CRIq_LA group (see Table 4 for further details).

The one-way ANOVA revealed a significant group effect on MMSE (F_1,72_ = 17.8; *p* = 0.0001; 23.08 ± 0.8 vs. 26.56 ± 0.41), SDMT (F_1,73_ = 13.1; *p* = 0.001; 15.5 ± 1.81 vs. 27.12 ± 2.02), VFT (F_1,73_ = 8.36; *p* = 0.005; 10.29 ± 0.74 vs. 13.58 ± 0.69), SCR (F_1,73_ = 6.63; *p* = 0.012; 36.75 ± 3.81 vs. 49.17 ± 2.78) and SWR (F_1,73_ = 10.87; *p* = 0.002; 48.42 ± 4.58 vs. 69.16 ± 3.73) at 2 years follow-up, showing poorer performance in the group with low CRIq_LA with respect to high CRI_LA (see Table 4 for further details).

*CRIq Total*: No statistically significant differences between groups were observed in any of the clinical and cognitive variables (see Appendix A).

*CRIq Education*: No statistically significant differences between groups were observed in any of the clinical and cognitive variables (see Appendix A).

*CRIq Working activity*: No statistically significant differences between groups were observed in any of the clinical and cognitive variables considered (see Appendix A).

CRIq_Edu: Cognitive Reserve Index education; CRIq_WA: Cognitive Reserve Index Working Activities; CRIq_LA: Cognitive Reserve Index Leisure Activities; CRIq_TOT: Cognitive Reserve Index Total Score.

∆TFC: the difference between the Total Functional Capacity score at t2 and t0; ∆TMS: the difference between the Total Motor Score at t2 and t0.

cUHDRS_t0: composite Unified Huntington’s Disease Rating Scale at baseline; cUHDRS_t1: composite UHDRS at 1-year follow-up; cUHDRS_t2: composite UHDRS at 2-year follow-up.

CRIq_Edu: Cognitive Reserve Index education; CRIq_WA: Cognitive Reserve Index Working Activities; CRIq_LA: Cognitive Reserve Index Leisure Activities; CRIq_Tot: Cognitive Reserve Index Total Score.

MMSE: Mini Mental State Examination; SDMT: Symbol Digit Modality Test; VFT: Categorical Verbal Fluency Test; SCR: Stroop Color Reading Test; SWR: Stroop Word Reading Test.

CRIq_LA: Cognitive Reserve Index Leisure Activities; ∆TFC: the difference between the Total Functional Capacity score at t2 and t0; ∆TMS: the difference between the Total Motor Score at t2 and t0.

cUHDRS_t0: composite Unified Huntington’s Disease Rating Scale at baseline; cUHDRS_t1: composite UHDRS at 1-year follow-up; cUHDRS_t2: composite UHDRS at 2-year follow-up.

MMSE: Mini Mental State Examination; SDMT: Symbol Digit Modality Test; VFT: Categorical Verbal Fluency Test; SCR: Stroop Color Reading Test; SWR: Stroop Word Reading Test.

SE: standard error; NS: not significant; t0: baseline; t1: 1 year follow-up; t2: 2 years follow-up.

CRIq_LA: Cognitive Reserve Index Leisure Activities; ∆TFC: the difference between the Total Functional Capacity score at t2 and t0; ∆TMS: the difference between the Total Motor Score at t2 and t0.

CRIq_LA: Cognitive Reserve Index Leisure Activities; MMSE: Mini Mental State Examination; SDMT: Symbol Digit Modality Test; VFT: Categorical Verbal Fluency Test; SCR: Stroop Color Reading Test; SWR: Stroop Word Reading Test; t0: baseline; t1: 1 year follow-up; t2: 2 years follow-up.

## 4. Discussion

The lifetime intellectual enrichment may significantly improve the cognitive and clinical performances in several neurological conditions [7]. Growing evidence is in favor of a positive influence of environment and cognition on the course of neurodegenerative diseases, including HD. For instance, bilingualism seems to exert a neuroprotective effect by influencing structural and metabolic brain changes [34]. Moreover, functional compensation is leveraged by lifestyle, i.e., the enhancement of physical and mental activities in AD and HD [34,35]. Additional examples concern the effect of educational level [25] and of biological factors [18,36] on neural compensation and eventually on clinical decline in HD.

One general limitation of studies on CR in HD is either a limited sample size or difficulty in analyzing specific tasks [24]. However, in an attempt to overcome this limitation, in this study we used the CRIq, an international questionnaire composed of several subscales, which has been used in normal and elderly populations and in other pathological conditions [37]. Among its subscales, we found the leisure time score exerted a major significant effect on clinical outcomes. Interestingly, we found that a higher leisure time was correlated with lesser cognitive and functional HD decline over time in early manifest HD subjects. Our findings are in line with the documented evidence of the beneficial effect of leisure activities on cognitive decline in the early stages of non-HD neurodegenerative diseases [38]. Leisure activities, that is, the time when a person is not working and can relax and do things that they enjoy (i.e., reading, theater, television, driving, internet, gardening, music etc.) contribute to increment a specific cognitive reserve that modulates the negative effect of brain pathology on cognition and independence in daily life. On the contrary, no effect of leisure time was observed on clinical motor variables (i.e., TMS); this is not surprising, as the cognitive reserve is mainly developed by exercising mental capacity, which seems to influence mainly the cognitive and functional domains [39].

HD patients with a higher reserve are able to withstand a more severe neuropathological burden; this protective effect of CR is shown by greater cognitive functioning as well as by less impairment in daily living activities. We found a long-lasting beneficial effect of CR on cognition and clinical outcomes that is consistent with previous CR research in AD and MS [40,41], and which could represent a potential neuroprotective target in HD as well.

We found that CRIq_Tot subscore, which includes education and work in addition to leisure activities, correlated with some cognitive measures. Specifically, CRIq_Tot correlated with MMSE over time, a screening measure of global cognitive impairment, as well as with SDMT at baseline. These results suggest a possible, although marginal influence of other CR variables on global cognitive deterioration over time.

Recent studies in HD show the existence of several neural mechanisms of compensation, suggesting the presence of a spread neural network to support cognitive dysfunctions; for a review, see [24,28]. High cognitive reserve seems to be associated with a reduced compensatory effect. Specifically, subjects who are in the early HD stages and experience an active cognitive lifestyle show reduced volume loss and reduced compensation in connectivity strength in specific brain regions related to executive control networks with respect to subjects with poorer cognitive lifestyle engagement [22,23]. These data are in accordance with the neural efficiency hypothesis [42]; people with a higher reserve require fewer cerebral resources to perform the same cognitive tasks than others with a lesser reserve.

Our study supports the hypothesis that at the beginning of the disease, leisure time plays a central role in preventing cognitive and functional degeneration over time when compared with education and work. A possible explanation is that the cognitive reserve provides constant exercise of the mental capacity during the whole life span. Conversely, it is possible that education may exercise this capacity only in the first part of life, while work may lose its power to influence this capacity over time as routines take over. After the formal education ends and work becomes more regular, leisurely cognitive activities provide ongoing mental exercise and stimulation that is critical for further development and maintenance of the cognitive reserve. Therefore, CR is not fixed over time, and it is modifiable in later life. This is supported by the evidence that cognitive stimulation and physical exercise increase neuroplasticity and promote neurogenesis [35]. The main biological mechanism underlying neural plasticity has been postulated to be an increase in BDNF levels. BDNF mediates the effect on brain engagement and cognitive reserve driven by physical exercise or by brain stimulation [43]. Modulating the cognitive reserve through lifestyle changes and/or through development of appropriate cognitive stimulation may represent a potential new avenue to counteract neurodysfunction and neurodegeneration.

Our findings highlight a negative correlation between mutation size and measured CR, thus supporting the interesting hypothesis that a larger expanded CAG repeat number in *HTT* gene is correlated with a lower patient CR. Even though our study did not reveal any mutation size amplification of the CRIq_LA effect on clinical measures, our data suggest a combinational environmental and genetic influence on the clinical development of the disease. The missing effect of the mutation on the relationship between CRIq_LA and the clinical measures may be explained by the relatively homogeneous CAG repeat number and by the limited range of expansions in our population. Larger studies including cohorts carrying higher repeat expansions (i.e., large mutations beyond 50 CAG repeats affecting young patients) may eventually highlight such a neurobiological effect of the mutation.

However, our study has limitations. First, even if we enrolled patients without severe psychiatric manifestations, we cannot fully exclude a possible influence of depression or anxiety on cognition. Second, the observational design of our study did not allow us to make causal inferences into the correlations between leisure time and the cognitive and functional domains.

Despite these limitations, our findings support the interesting hypothesis that in addition to possible genetic modifiers, the environment may consistently contribute to disease course modification and may represent a therapeutic target. Further studies on larger cohorts investigating the role of leisure time in HD are required in order to confirm our hypothesis and to address conclusions.

## 5. Conclusions

In conclusion, our results highlighting a combined effect of environmental and neurobiological factors (i.e., the expanded CAG repeat number in *HTT* gene) on the development of HD suggest that the CR may be affected in HD, and that increased leisure time lowers HD progression as measured by functional and multidimensional variables, with more preserved cognitive status over time. These aspects could be considered in clinical management to address HD by directing patients to specific cognitive training programs and clinical trial recruitments in order to minimize potential selection bias.

## Figures and Tables

**Figure 1 jpm-12-00036-f001:**
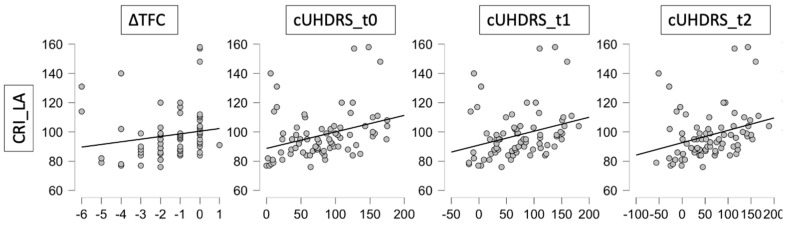
Correlation Matrix plots between clinical variables and Cognitive Reserve Index Leisure Activities.

**Figure 2 jpm-12-00036-f002:**
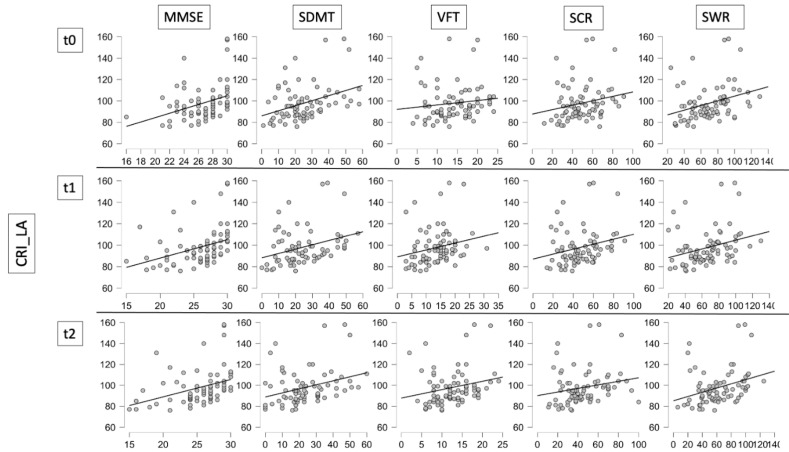
Correlation Matrix plots between cognitive variables and Cognitive Reserve Index Leisure Activities.

**Table 1 jpm-12-00036-t001:** Clinical and demographic characteristics of the study sample.

Early Manifest HD Cohort	
Sample Number	75
Male-Female	47-28
Age at baseline (range)	47.2 ± 12.5 (27–78)
Education level in years (range)	11.65 ± 4.6 (8–18)
Age at motor onset (range)	47.5 ± 12.3 (22–70)
CAG repeat number (range)	43.7 ± 2.3 (40–49)

**Table 2 jpm-12-00036-t002:** Pearson’s *r* (and related level of significance) between Cognitive Reserve Index subscores and clinical variables.

	Age of Onset		∆TFC		∆TMS		cUHDRS_t0		cUHDRS_t1		cUHDRS_t2	
	*r*	*p*	*r*	*p*	*r*	*p*	*r*	*p*	*r*	*p*	*r*	*p*
CRIq_Edu	0.156	0.181	0.036	0.758	−0.104	0.375	0.049	0.690	0.029	0.805	0.018	0.878
CRIq_WA	0.026	0.824	0.058	0.619	−0.141	0.228	0.058	0.624	0.078	0.508	0.065	0.580
CRIq_LA	0.095	0.418	0.242	0.037	−0.150	0.200	0.349	0.002	0.332	0.004	0.344	0.003
CRIq_Tot	0.170	0.146	0.117	0.316	−0.117	0.319	0.191	0.100	0.179	0.124	0.149	0.203

**Table 3 jpm-12-00036-t003:** Pearson’s *r* (and related level of significance) between Cognitive Reserve Index subscores and cognitive variables: (**a**) Pearson’s *r* (and related level of significance) between Cognitive Reserve Index subscores and MMSE, SDMT, VTF, SCR and SWR at baseline (t0); (**b**) Pearson’s *r* (and related level of significance) between Cognitive Reserve Index subscores and MMSE, SDMT, VTF, SCR and SWR at 1 year follow-up (t1); (**c**) Pearson’s *r* (and related level of significance) between Cognitive Reserve Index subscores and MMSE, SDMT, VTF, SCR and SWR at two-year follow-up (t2).

(**a**)										
	**MMSE**		**SDMT**		**VFT**		**SCR**		**SWR**	
	** *r* **	** *p* **	** *r* **	** *p* **	** *r* **	** *p* **	** *r* **	** *p* **	** *r* **	** *p* **
CRIq_Edu	0.206	0.082	0.026	0.825	0.080	0.494	0.019	0.870	0.117	0.319
CRIq_WA	0.146	0.220	0.137	0.242	−0.160	0.171	−0.016	0.892	0.079	0.500
CRIq_LA	0.340	0.003	0.392	0.001	0.240	0.036	0.280	0.015	0.348	0.002
CRIq_Tot	0.294	0.012	0.239	0.039	0.025	0.831	0.100	0.393	0.220	0.058
(**b**)										
	**MMSE**		**SDMT**		**VFT**		**SCR**		**SWR**	
	** *r* **	** *p* **	** *r* **	** *p* **	** *r* **	** *p* **	** *r* **	** *p* **	** *r* **	** *p* **
CRIq_Edu	0.205	0.083	−0.005	0.970	0.106	0.365	0.071	0.547	0.075	0.522
CRIq_WA	0.192	0.098	0.114	0.337	0.027	0.820	0.064	0.588	0.082	0.485
CRIq_LA	0.396	<0.001	0.359	0.002	0.265	0.022	0.294	0.010	0.330	0.004
CRIq_Tot	0.337	0.003	0.192	0.103	0.156	0.183	0.171	0.142	0.200	0.085
(**c**)										
	**MMSE**		**SDMT**		**VFT**		**SCR**		**SWR**	
	** *r* **	** *p* **	** *r* **	** *p* **	** *r* **	** *p* **	** *r* **	** *p* **	** *r* **	** *p* **
CRIq_Edu	0.193	0.100	−0.021	0.860	0.082	0.486	−0.001	0.991	0.046	0.694
CRIq_WA	0.205	0.080	0.087	0.456	0.069	0.554	0.037	0.750	0.079	0.499
CRIq_LA	0.404	<0.001	0.354	0.002	0.252	0.029	0.253	0.028	0.374	0.001
CRIq_Tot	0.326	0.005	0.178	0.128	0.167	0.153	0.113	0.334	0.200	0.085

**Table 4 jpm-12-00036-t004:** Clinical and cognitive scores in impaired and normal cognitive reserve leisure activities groups.

		CRI_LA Impaired Group (*n* = 24) Mean ± SE	CRI_LA Normal Group (*n* = 51) Mean ± SE	*p*
Clinical variables	∆_TFC	−2.08 ± 0.29	−1.2 ± 0.21	0.017
	∆_TMS	9.79 ± 1.94	8.25 ± 1.24	0.496
	cUHDRS_t0	54.34 ± 8.23	92.28 ± 6.69	0.001
	cUHDRS_t1	41.09 ± 8.41	85.48 ± 7.14	<0.001
	cUHDRS_t2	30.09 ± 9.64	75.50 ± 8.13	<0.001
Cognitive variables - Baseline	MMSE	25.34 ± 0.60	26.98 ± 0.37	0.019
	SDMT	17.54 ± 1.70	28.58 ± 1.92	0.001
	VFT	11.87 ± 0.84	14.94 ± 0.75	0.016
	SCR	41.41 ± 3.19	52.88 ± 2.42	0.007
	SWR	58.75 ± 4.39	75.02 ± 3.35	0.006
Cognitive variables - 1 year follow-up	MMSE	23.62 ± 0.74	26.78 ± 0.41	<0.001
	SDMT	16.04 ± 1.84	28.06 ± 1.91	<0.001
	VFT	10.7 ± 1	14.94 ± 0.8	0.003
	SCR	39.25 ± 2.71	50.17 ± 2.63	0.013
	SWR	54.16 ± 3.92	72.6 ± 3.49	0.002
Cognitive variables - 2 years follow-up	MMSE	23.08 ± 0.8	26.56 ± 0.41	<0.001
	SDMT	15.5 ± 1.81	27.12 ± 2.02	0.001
	VFT	10.29 ± 0.74	13.58 ± 0.69	0.005
	SCR	36.75 ± 3.81	49.17 ± 2.78	0.012
	SWR	48.42 ± 4.58	69.16± 3.73	0.002

## Data Availability

The data that support the findings of this study are available from the corresponding author upon reasonable request.

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
