# Peer review of "Cognitive Reserve in Early Manifest Huntington Disease Patients: Leisure Time Is Associated with Lower Cognitive and Functional Impairment"

_jpm, 2022, doi:10.3390/jpm12010036_

Round 1

Reviewer 1 Report

Thanks for recommending me as a reviewer. In this paper, the authors focused on cognitive reserve (CR) in patients with early Huntington's disease (HD) and investigated whether clinical outcomes could be affected by lifetime intellectual enrichment, overtime. If the authors complete the revision, the quality of the study will be further improved.

  1. The introduction section is well written. If the authors describe in more detail the trends in prior research related to cognitive reserve in the early manifest Huntington disease patients in the introduction section, it can help readers understand.
  2. line 100: Patients’ population - Authors should be more specific about the subject, such as sampling.
  3. line 100: Is the subheading “patients’ population” appropriate? This study is not an epidemiologic study. In my opinion, subjects are more suitable as subheadings.
  4. line 134: Cognitive reserve - Authors should be more specific about CRIq.
  5. Table3b: Significance level of 0.000 should be changed to p<0.001.
  6. Table4: The number of decimal places should be the same in indications such as significance level.
  7. It is a good idea to separate the last paragraph of the discussion section into a conclusion section.

Author Response

1. REPLY: We thank the reviewer for feedback. We updated the Introduction as suggested and added three more references.

2. REPLY: We have now improved the patients’ description in Methods. Please see also Table 1 for further details.

3. REPLY: Amended

4. REPLY: We added more information in the dedicated section (lines 168-169, lines 172-173, lines 230-231)

5. REPLY: Amended

6. REPLY: Amended

7. REPLY: Amended

Reviewer 2 Report

Thank you for giving me the opportunity to review this interesting paper on the relationship between cognitive reserve and cognitive decline and functional impairment in Huntington's disease. The the study design is sound and the results well-presented and of interest for the journal's readership. However, I believe a number of minor points should be address prior to publication:

  1. The recruitment strategy would benefit from further clarification - e.g., a brief description of what Enroll-HD is and the timeframe in which the recruitment took place. 
  2. The title of the study claims that leisure time 'mitigates' cognitive decline and functional impairment. This implies a causal effect. However, the paper reports an observational design based on a convenience sample, and the findings are mainly based on correlational analyses. As such, no insight into causation can be inferred, and I suggest rewording the title to reflect the correlational nature of the results (e.g., leisure time 'is associated with' cognitive decline and functional impairment).
  3. Based on Point 2, the observational design and the convenience sample should be added as limitations in the discussion. 
  4. Was the participants' mental health status (e.g., levels of anxiety and depression) checked and considered during the analysis? If not, this should also be added as a limitation, as poor mental health can have significant effects on cognition and cognitive reserve, and in particular engagement with leisure activities. 

Author Response

  1. REPLY: Amended. Please, see Methods’ section.
  2. REPLY: We thank the reviewer and changed the title accordingly.

  3. REPLY: We added one more sentence on study limitation in Discussion section, from line 518 to line 522.

  4. REPLY: We excluded patients with severe psychiatric manifestations (see Methods section). We did not include the patients’ mental health status in our analyses. Therefore, we added this limitation in Discussion, as suggested.

Round 2

Reviewer 1 Report

The authors faithfully completed the revision.

This manuscript is a resubmission of an earlier submission. The following is a list of the peer review reports and author responses from that submission.

Round 1

Reviewer 1 Report

In this paper, the authors focused on Cognitive Reserve (CR) in patients with early Huntington Disease (HD) and investigated whether clinical outcomes may be influenced by the lifetime intellectual enrichment, overtime. If the authors complete the revision, the quality of the study will be further improved.

  1.  The introduction section is well written. It may be helpful for readers to understand if the authors describe the research trends in the introductory section more specifically about the relationship between leisure time (or lifestyle) and cognitive impairment (or cognitive function).

2. I suggest that authors combine some paragraphs of the introduction with others. For example, lines 56-59 can be combined with the lines 60-69 paragraphs.

3. line 137-143: "At first, we conducted an exploratory Pearson’s correlation analysis in order to detect potential relationships between CRIq scores, (CRIq_Tot, CRIq_Edu, CRIq_WA, CRIq_LA) and the clinical and cognitive variables under investigation [Age at Onset (AO), ∆TMS (the difference between the TMS score at t2 and t0), ∆TFC, cUHDRS, cognitive variables (VFT, correct response; SCR, correct responses; SWR, correct responses; SDMT, correct responses) and MMSE]. The correlation analysis was performed for all time points (t0, t1, and t2). " - For testing tools such as MMSE, references should be indicated along with the full name.

4. line 130: http://cri.psy.unipd.it/   - Thank you for providing your site address. But I couldn't access this web address. Authors need to verify that access to the web page is valid.

5. line 131: "Ethical approval" - The Ethical approval section is recommended to go to the “Patients’ population” section.  Because readers will be curious about ethical issues when reading the Patients’ population section.

6. Table 2, 3, 4...: When indicating the significance level in a table, it is more common to indicate a separate footnote (*0.05<p).

7. Unlike the previous table, Table 4 did not emphasize the significance level.

8. Authors should add limitations to the discussion section.

Author Response

Reviewer #1

In this paper, the authors focused on Cognitive Reserve (CR) in patients with early Huntington Disease (HD) and investigated whether clinical outcomes may be influenced by the lifetime intellectual enrichment, overtime. If the authors complete the revision, the quality of the study will be further improved.

  1. The introduction section is well written. It may be helpful for readers to understand if the authors describe the research trends in the introductory section more specifically about the relationship between leisure time (or lifestyle) and cognitive impairment (or cognitive function).

REPLY: We added two sentences in the introduction section, as suggested by the reviewer.

  1. I suggest that authors combine some paragraphs of the introduction with others. For example, lines 56-59 can be combined with the lines 60-69 paragraphs.

REPLY: Amended

  1. line 137-143: "At first, we conducted an exploratory Pearson’s correlation analysis in order to detect potential relationships between CRIq scores, (CRIq_Tot, CRIq_Edu, CRIq_WA, CRIq_LA) and the clinical and cognitive variables under investigation [Age at Onset (AO), ∆TMS (the difference between the TMS score at t2 and t0), ∆TFC, cUHDRS, cognitive variables (VFT, correct response; SCR, correct responses; SWR, correct responses; SDMT, correct responses) and MMSE]. The correlation analysis was performed for all time points (t0, t1, and t2). " - For testing tools such as MMSE, references should be indicated along with the full name.

REPLY: Amended

  1. line 130: http://cri.psy.unipd.it/   - Thank you for providing your site address. But I couldn't access this web address. Authors need to verify that access to the web page is valid.

REPLY: We thank the reviewer for highlighting this and we have now replaced the correct website address in the text (https://dpg.unipd.it/en/criq)

  1. line 131: "Ethical approval" - The Ethical approval section is recommended to go to the “Patients’ population” section.  Because readers will be curious about ethical issues when reading the Patients’ population section.

REPLY: We moved the Ethical approval to the “Patients’ population” section.

  1. Table 2, 3, 4...: When indicating the significance level in a table, it is more common to indicate a separate footnote (*0.05<p).

REPLY: We thank the reviewer for this suggestion. However, we preferred to include the real p numbers in the table instead of asterisks in a graphic figure.

  1. Unlike the previous table, Table 4 did not emphasize the significance level.

REPLY: We added the missing significance level to ∆_TMS.

  1. Authors should add limitations to the discussion section.

REPLY: We added limitations to the last paragraph of Discussion

Reviewer 2 Report

The study is not suitable for drawing conclusions from the current number of patients.

The number of samples is extremely insufficient. I also think that parameters such as scale and survey are not strong enough for these inferences.

Author Response

Reviewer #2

The study is not suitable for drawing conclusions from the current number of patients.

Reply: We agree with the reviewer even though our cohort of 75 early manifest patients, out of 177 selected in origin, does anyway represent one of the largest HD populations ever described in studies on cognitive reserve. Indeed, we only suggest hypotheses in our manuscript and did not draw final conclusions yet (see Discussion).

The number of samples is extremely insufficient.

Reply: Large cohorts may only be recruited by huge observational programs such as Enroll-HD, which we take part in. However, large observational programs do not generally include specific tasks that are of critical importance to explore new hypotheses. Indeed, the task we investigated in our current work is not included in that platform.

I also think that parameters such as scale and survey are not strong enough for these inferences.

Reply: This is in line with a quite recent Review, which we have now added to references, from Soloveva et al on cognitive reserve in HD (Neurosci Biobehav Rev 2018 May;88:155-169), whose conclusions was that the 'reserve' hypothesis should be tested only empirically and that it is important to identify and embed potential neuroprotective modulating factors of 'reserve'. We are in line with such recommendations/conclusions and we believe that the investigation of the leisure time task may contribute to explore new potential modulating factors thus opening the door to further studies on larger cohorts in future. We also agree with the reviewer’s criticism that the study design should be improved and we believe that our hypothesis taking into account the role of the leisure time could represent the basis to develop new investigational protocols in future.

Round 2

Reviewer 1 Report

The authors faithfully completed the revision.

Reviewer 2 Report

This limited number of patients, which the authors advocate as a "hypothesis", has the potential to be taken as a basis in future cases, as they themselves stated. In order to give a more accurate direction to these studies, either a method with more concrete data than the questionnaire should be used or a larger number of patients should be examined. The study as such is not suitable for publication in this journal.